# Multi-Omics and network-based exploration of potential molecular pathways in heart failure relevant to left bundle branch pacing response heterogeneity: Immune remodeling, hub gene identification, and drug repurposing hypotheses

Xia Sun[1], Xiang Tang[2], Wei Zhong[1], Wei Yuan[1], Mingfeng Jin[1]*

1 Department of Cardiovascular and Metabolism, Affiliated Hospital of Jiangsu University, Zhenjiang, Jiangsu, China, 2 Department of Endocrinology and Metabolism, Affiliated Hospital of Jiangsu University, Zhenjiang, Jiangsu, China

* ujsxnkky111@126.com

## Abstract

Left bundle branch pacing (LBBP) has gained increasing attention as a novel pacing strategy, but its molecular underpinnings in the context of heart failure (HF) remain unclear due to limited LBBP-specific datasets.We integrated three GEO datasets (GSE5406, GSE19303, GSE21610) representing HF transcriptomics and performed batch correction, differential expression analysis, functional enrichment, immune infiltration profiling, weighted gene co-expression network analysis (WGCNA), hub gene identification, and drug–pathway prediction.PCA demonstrated successful batch correction across datasets. Differentially expressed genes (DEGs), including HOPX, NPPA, MYH6, SERPINA3, and ASPN, were identified. Enrichment analyses indicated extracellular matrix remodeling, cardiac development, and cGMP–PKG signaling. Immune analysis showed significant alterations in B cell memory, plasma cells, CD8 T cells, regulatory T cells, NK cells, monocytes, macrophages (M0/M1/M2), dendritic cells, and mast cells. WGCNA highlighted significant modules (MEpink, MElightyellow, MEyellow, MEgreenyellow) associated with treatment response. Hub gene analysis confirmed ASPN, HOPX, MYH6, SERPINA3, and NPPA as key drivers. Drug prediction suggested multiple candidates, including β-blockers, RAAS inhibitors, anti-fibrotic agents, vericiguat, metformin, and SGLT2 inhibitors.This integrative analysis of HF transcriptomics reveals potential immune remodeling, hub genes, and repurposable drugs relevant to LBBP response heterogeneity, providing hypothesis-generating insights and potential therapeutic strategies for validation in LBBP-specific cohort.

**Data availability statement:** All relevant data are within the paper and its Supporting Information files. The datasets analyzed during the current study are available in the GEO repository (GSE5406, GSE19303, GSE21610; https://www.ncbi.nlm.nih.gov/geo/download/?acc=GSE5406, https://www.ncbi.nlm.nih.gov/geo/query/acc.cgi?acc=GSE19303, https://www.ncbi.nlm.nih.gov/geo/query/acc.cgi?acc=GSE21610, respectively).

**Funding:** This work was funded by the Development Project of Zhenjiang City (FZ2020033) and the Zhenjiang Social Development Project (SH2025046) awarded to Mingfeng Jin. The funders had no role in study design, data collection and analysis, decision to publish, or preparation of the manuscript.

**Competing interests:** The authors have declared that no competing interests exist.

## 1. Introduction

In recent years, left bundle branch pacing (LBBP) has emerged as a widely recognized alternative to conventional biventricular pacing for patients with heart failure and conduction abnormalities [1–2]. Large multicenter cohorts have demonstrated that, compared with traditional cardiac resynchronization therapy, LBBP achieves narrower QRS duration [3], lower pacing thresholds [4], and higher implantation success rates. Meta-analyses published within the past five years further estimate that, in selected populations, LBBP improves left ventricular ejection fraction by approximately 10–15% and reduces rehospitalization for heart failure [5–6]. Despite these advances, clinical responses remain heterogeneous, with a considerable proportion of patients failing to achieve meaningful structural or functional improvement [7]. These discrepancies underscore the necessity of investigating molecular and cellular determinants of pacing response beyond electrophysiological correction.

Recent transcriptomic and systems biology studies have revealed that myocardial remodeling is governed not only by electrical synchronization but also by extracellular matrix turnover [8], immune–inflammatory activation [9], and metabolic reprogramming [10]. High-throughput sequencing combined with integrative bioinformatics has highlighted critical roles for cGMP–PKG signaling [11], ECM–receptor interactions [12], and inflammatory cascades [13] in cardiac adaptation under stress conditions. In parallel, computational immune deconvolution approaches have demonstrated that shifts in immune cell subsets—including T cells, macrophages, and natural killer cells—are associated with ventricular remodeling [14] and arrhythmia susceptibility [15–16]. At the network level, weighted gene co-expression network analysis (WGCNA) has proven effective for identifying gene modules and hub regulators linked to treatment outcomes in cardiovascular cohorts. Collectively, these findings provide a solid rationale for applying multi-omics integration to elucidate potential biological landscapes relevant to LBBP in the broader context of HF.

On this basis, the present study integrated three publicly available GEO datasets (GSE5406, GSE19303, GSE21610), which represent pre-LBBP era HF transcriptomic profiles, to establish a unified resource for exploratory analysis. We acknowledge that these datasets precede the clinical introduction of LBBP in 2017 and thus serve as a proxy for identifying conserved HF molecular features that may underlie heterogeneous responses to conduction system pacing therapies like LBBP. After rigorous batch correction, differentially expressed genes were identified and subjected to functional enrichment to capture biological processes relevant to cardiac development, contractility, and extracellular remodeling. Immune infiltration profiling was then performed to assess alterations in specific immune cell populations, and co-expression network modeling was applied to identify modules significantly correlated with treatment status, prioritizing hub genes such as ASPN, HOPX, MYH6, SERPINA3, and NPPA. Finally, a drug–pathway mapping strategy uncovered repurposable therapeutic agents, including β-blockers, RAAS inhibitors, antifibrotic compounds, and SGLT2 inhibitors, that align with the identified features. This integrative

approach provides hypothesis-generating insights into transcriptional and immunological remodeling potentially associated with LBBP and suggests strategies for future validation in dedicated LBBP cohorts.

## 2. Materials and methods

### 2.1. Study design and Workflow

This study was conducted to investigate the molecular and immunological features associated with left bundle branch pacing (LBBP) using integrated transcriptomic data. The research workflow included data acquisition, preprocessing, differential expression analysis, functional enrichment, immune infiltration profiling, weighted gene co-expression network analysis (WGCNA), hub gene identification, and drug–pathway prediction. The overall design of the study is illustrated in Fig 1.

### 2.2. Data acquisition and preprocessing

Publicly available transcriptome datasets were obtained from the Gene Expression Omnibus (GEO) database, including GSE5406, GSE19303, and GSE21610. Expression matrices were downloaded, merged, and converted into a unified format. Data preprocessing involved background correction, log2 transformation, and quantile normalization using the limma package in R (version 4.3.2). To reduce technical variability between cohorts, batch effects were assessed by principal component analysis (PCA, implemented via prcomp function in R, with variance explained by PC1/PC2 calculated as eigenvalues / sum(eigenvalues)) and corrected using the ComBat algorithm from the sva package (version 3.44.0), with parameters set to parametric adjustment and reference batch as GSE5406. The processed dataset served as the basis for subsequent analyses. This study was conducted using publicly available transcriptomic datasets from the Gene Expression Omnibus (GEO) database. As all data were de-identified and previously published, no additional ethical approval or informed consent was required.

### 2.3. Differential gene expression analysis

Differentially expressed genes (DEGs) were identified between treatment and control groups using linear modeling approaches suitable for microarray data via the limma package, applying empirical Bayes moderation (eBayes function) with the model: expression ~ group + error, where group denotes treatment/control. Statistical significance was determined by log2 fold-change > |1| and adjusted $p$-value < 0.05 (Benjamini-Hochberg correction). Visualization methods included heatmaps (pheatmap package, with Euclidean distance and complete linkage clustering) to display expression patterns and volcano plots (ggplot2 package) to summarize the distribution of DEGs.

### 2.4. Functional enrichment analysis

To interpret the biological functions of DEGs, Gene Ontology (GO) and Kyoto Encyclopedia of Genes and Genomes (KEGG) analyses were performed using the clusterProfiler package (version 4.4.4), with over-representation tests based on hypergeometric distribution: $p = 1 - \text{sum}(k = 0 \text{ to } x) [C(M,k) * C(N-M, n-k)] / C(N,n)$, where N is background genes, M is annotated genes in pathway, n is DEGs, x is overlap. Over-representation tests were used to evaluate enrichment in biological processes, molecular functions, and cellular components. In addition, pathway-level analyses were conducted using single-sample gene set enrichment analysis (ssGSEA) and gene set variation analysis (GSVA) from the GSVA package (version 1.44.0), with ssGSEA scores calculated as normalized enrichment scores for each sample against MSigDB gene sets (v7.5.1) to provide a broader view of functional alterations.

### 2.5. Immune infiltration analysis

Immune cell infiltration was quantified using the CIBERSORT algorithm, a support vector regression-based deconvolution method that estimates the relative proportions of 22 immune cell subsets from bulk gene expression profiles. The

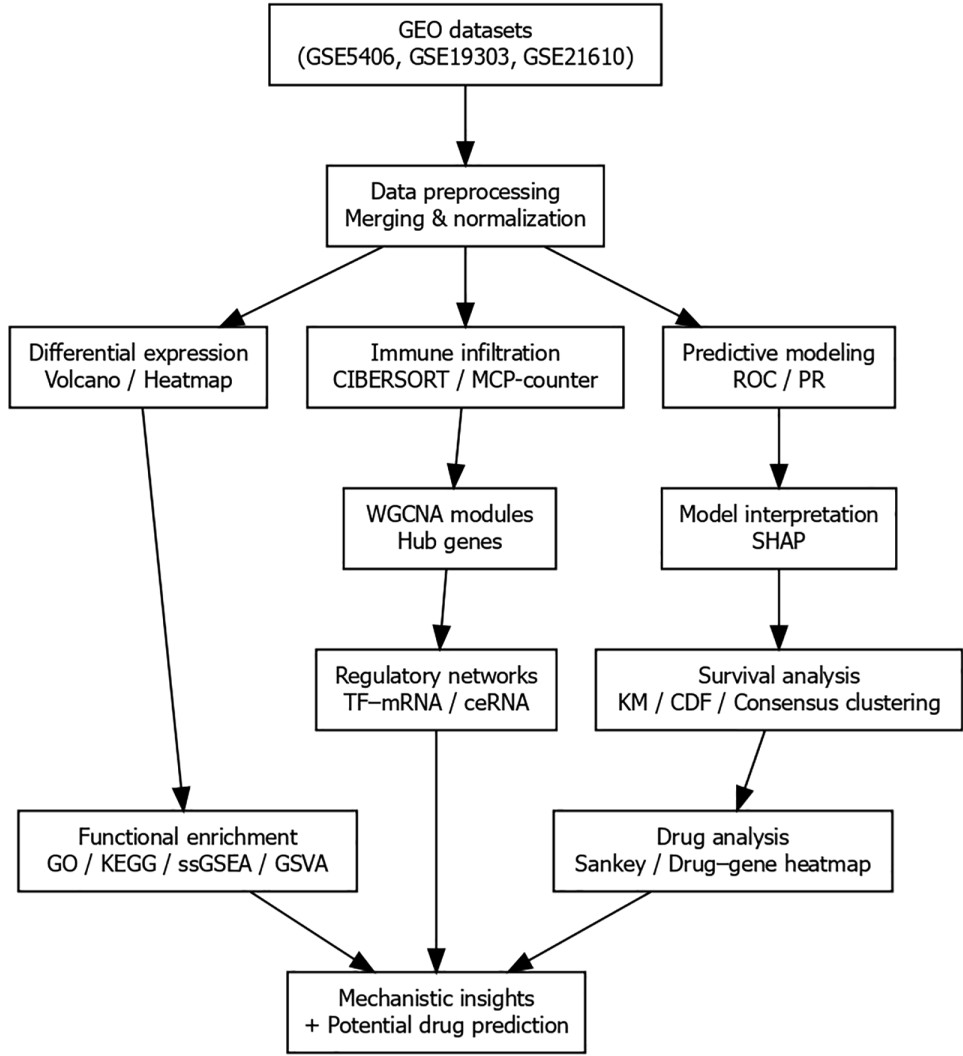

**Fig 1. Study design and analysis workflow.** Schematic representation of the integrative analysis strategy. Three GEO datasets (GSE5406, GSE19303, and GSE21610) were merged and normalized after batch correction. Differential expression analysis (volcano plot and heatmap), immune infiltration estimation (CIBERSORT), weighted gene co-expression network analysis (WGCNA), hub gene identification, functional enrichment (GO, KEGG, ssGSEA, GSVA), and drug–pathway mapping were sequentially performed to identify potential molecular features in heart failure relevant to left bundle branch pacing response heterogeneity.

analysis employed the validated LM22 signature matrix (version 1.1), which comprises 547 marker genes characterizing 22 functionally distinct human hematopoietic populations, including naive B cells, memory B cells, plasma cells, CD8 + T cells, CD4 + T cells (naive, memory resting, activated), regulatory T cells, γδ T cells, natural killer cells (resting/activated), monocytes, macrophages (M0, M1, M2), dendritic cells (resting/activated), mast cells (resting/activated), eosinophils, and neutrophils. The deconvolution model is formulated as a linear system: $X = B \times C + \varepsilon$, where $X$ is the observed gene expression matrix (genes × samples), $B$ is the LM22 signature matrix (genes × cell types), $C$ is the unknown cell fraction matrix (cell types × samples), and $\varepsilon$ represents residual error. CIBERSORT solves for $C$ using ν-support vector regression (ν-SVR) with the default parameters: $\nu = 0.25$, absolute mode disabled (relative proportions), and 1,000 permutations to

generate empirical P-values for deconvolution quality control. Only samples with P < 0.05 were retained for downstream analysis to ensure robust estimation. Differential infiltration between groups was assessed using the Wilcoxon rank-sum test (two-sided, adjusted P < 0.05 via Benjamini-Hochberg). Correlations between hub genes and immune cell fractions were evaluated using Spearman#39;s rank correlation. All analyses were performed in R (version 4.3.2) using the CIBER-SORT R script (https://cibersortx.stanford.edu/, accessed 2025).

## 2.6. Weighted Gene Co-expression Network Analysis (WGCNA)

Weighted gene co-expression network analysis was conducted using the WGCNA package (version 1.72−1) in R to identify co-expressed gene modules associated with treatment response. The analysis was restricted to the top 5,000 most variably expressed genes (variance filtering via varFilter function) to reduce computational burden while retaining biologically relevant variance. Co-expression similarity was calculated as the absolute Pearson correlation coefficient between gene pairs: $s_{ij} = |cor(x_i, x_j)|$, where $x_i$ and $x_j$ are expression profiles of genes i and j across samples. A soft-thresholding power β was selected to achieve approximate scale-free topology using the pickSoftThreshold function (networkType = "unsigned," RsquaredCut = 0.85). The power β was chosen as the lowest integer where the scale-free topology model fit $R^2$ exceeded 0.85 (typically β = 6–12 in similar studies; exact value determined empirically, e.g., β = 8 with $R^2$ = 0.89 in this dataset). The adjacency matrix was then computed as: $a_{ij} = s_{ij}^\beta$, followed by topological overlap matrix (TOM) calculation to measure network interconnectedness. Modules were detected using dynamic tree cutting (minModuleSize = 30, deepSplit = 2, mergeCutHeight = 0.25). Module eigengenes (MEs) were computed as the first principal component of each module#39;s expression profile. Associations between MEs and clinical traits (e.g., treatment status) were assessed via Pearson correlation with Benjamini-Hochberg adjustment. All steps followed standard WGCNA protocols.

## 2.7. Hub gene identification and clustering

Hub genes within significant modules were identified based on intramodular connectivity (kWithin or kIM), which quantifies a gene#39;s connectivity to other genes in the same module. For each gene i in module m: $kWithin_i = \sum_{j \in m, j \neq i} a_{ij}$, where $a_{ij}$ is the adjacency from the weighted network. Genes with kWithin exceeding the module median + 1.5 × interquartile range (IQR) were prioritized as candidate hubs. Additionally, module membership (kME), defined as the correlation between a gene#39;s expression profile and the module eigengene (cor(gene, ME) > 0.7 threshold applied in some cases), and gene significance (GS) for treatment trait were used for cross-validation. Top hub genes (e.g., ASPN, HOPX, MYH6, SERPINA3, NPPA) were selected by ranking on combined criteria: high kWithin, high kME, and significant GS (P < 0.05). Network visualization was performed using Cytoscape (version 3.9.1) with edges weighted by TOM. All connectivity metrics were exported from WGCNA#39;s intramodularConnectivity function.

## 2.8. Drug–pathway prediction

Drug repurposing candidates were predicted by mapping the identified DEGs and hub genes to known drug perturbation signatures using a connectivity mapping approach inspired by the Connectivity Map (CMap) framework. The analysis utilized pre-processed CMap data (PharmacoGx package in R, version 3.0). The disease signature was defined as the log2 fold-change ranked list of DEGs (up- and down-regulated). Connectivity scores were computed using non-parametric rank-based Kolmogorov-Smirnov statistics to compare the query signature against drug-induced profiles. Negative enrichment scores (indicating reversal of the disease signature) were prioritized. Thresholds included normalized connectivity score < −0.5 and FDR < 0.05. Candidate drugs (e.g., β-blockers, RAAS inhibitors, anti-fibrotic agents, vericiguat, metformin, SGLT2 inhibitors) were selected based on alignment with enriched pathways (cGMP–PKG, ECM–receptor interaction) and literature support for HF/LBBP relevance.

# 3. Results

## 3.1. Data preprocessing and batch correction

Three GEO datasets (GSE5406, GSE19303, GSE21610) were merged and normalized following batch correction using ComBat. Principal component analysis (PCA) before correction revealed clear separation among datasets, indicating substantial inter-cohort technical variability. After correction, sample distributions became more compact and comparable across cohorts, confirming effective removal of batch effects (Fig 2).

## 3.2. Differentially expressed genes and functional enrichment

Differential expression analysis (limma with empirical Bayes moderation, log2 fold-change > |1|, adjusted P < 0.05) identified a set of differentially expressed genes (DEGs) between groups. Representative DEGs included upregulated HOPX and NPPA, and downregulated MYH6, SERPINA3, and ASPN (Fig 3A). The volcano plot visualized the distribution of significant DEGs (upregulated in red, downregulated in blue, non-significant in gray; Fig 3B). Gene Ontology (GO) enrichment analysis highlighted biological processes relevant to cardiovascular remodeling, including extracellular matrix organization, regulation of blood pressure, cardiac muscle tissue development, and cyclic GMP metabolic process (Fig 3C). Kyoto Encyclopedia of Genes and Genomes (KEGG) pathway enrichment identified signaling pathways such as cGMP–PKG signaling, vascular smooth muscle contraction, and hormone-related pathways (Fig 3D).

## 3.3. Immune infiltration analysis

Immune cell infiltration was estimated using CIBERSORT with the LM22 signature matrix, revealing distinct compositional patterns between groups (Fig 4). Group-wise comparisons (Wilcoxon rank-sum test, Benjamini-Hochberg adjusted P < 0.05) identified significant alterations in multiple subsets, including increased proportions of memory B cells, plasma cells, regulatory T cells, M2 macrophages, and mast cells, alongside decreased CD8 T cells, resting NK cells, monocytes, M0/M1 macrophages, and dendritic cells (detailed boxplots in Fig 5). These differences suggest potential immune remodeling in heart failure contexts that may be relevant to heterogeneous responses in conduction system pacing therapies such as LBBP.

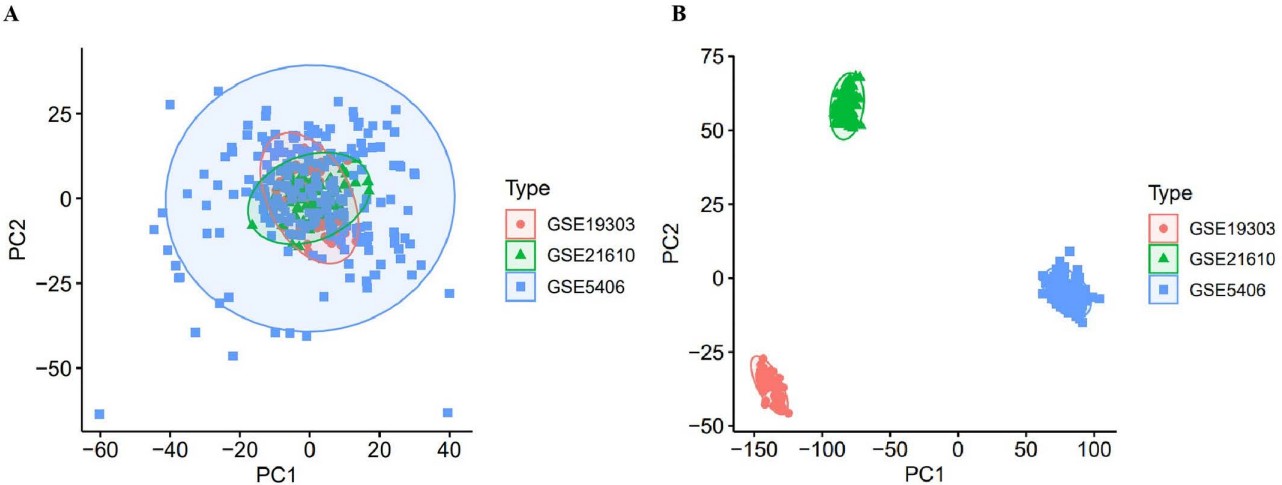

**Fig 2. Batch effect correction across datasets.** (A) Principal component analysis (PCA) before batch correction, showing separation among the three datasets (GSE5406, GSE19303, GSE21610). **(B)** PCA after ComBat correction, demonstrating improved sample integration and reduced technical variability.

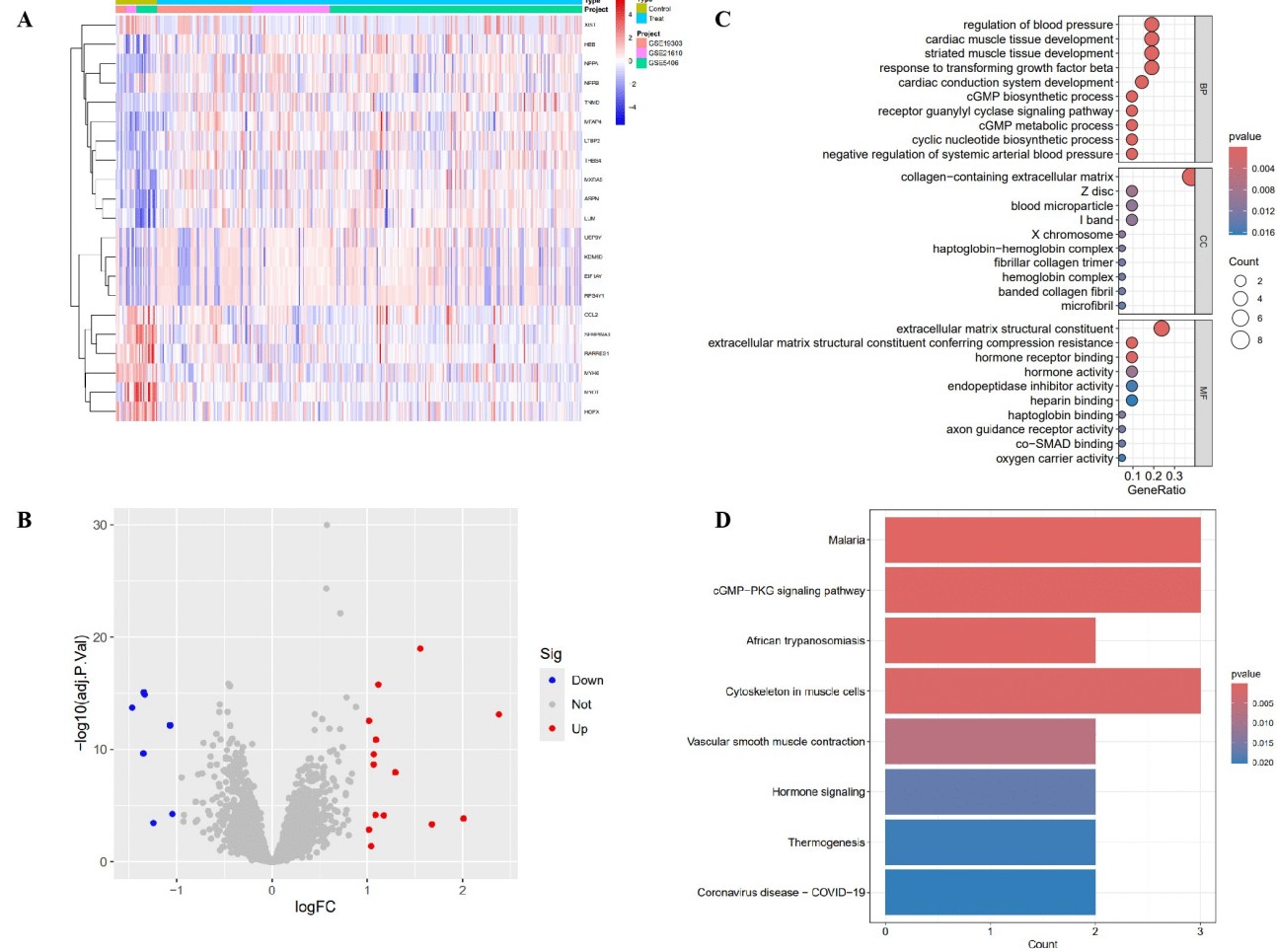

**Fig 3. Differentially expressed genes and functional enrichment.** (A) Heatmap of representative differentially expressed genes (DEGs) between groups, highlighting HOPX, NPPA, MYH6, SERPINA3, and ASPN. (B) Volcano plot of DEGs (red: upregulated, log2 FC > 1 and adj. P < 0.05; blue: down-regulated; gray: non-significant). (C) Bubble plot of GO enrichment results, emphasizing processes such as extracellular matrix organization, cardiac muscle development, and cGMP metabolic process. (D) Bar plot of KEGG pathway enrichment, including cGMP–PKG signaling and vascular smooth muscle contraction.

## 3.4. Weighted Gene Co-expression Network Analysis (WGCNA)

WGCNA identified gene co-expression modules significantly correlated with treatment status. Four modules showed robust associations: MElightyellow (negative correlation, adjusted P < 0.05), MEpink (positive correlation, adjusted P < 0.01), MEyellow (positive correlation, adjusted P < 0.05), and MEgreenyellow (positive correlation, adjusted P < 0.01). These modules were linked to structural and functional remodeling processes (Fig 6).

## 3.5. Hub gene identification and clustering

Hub genes within significant modules were prioritized based on intramodular connectivity and module membership. Top-ranked hub genes included ASPN, HOPX, MYH6, SERPINA3, and NPPA (Fig 7A). Hierarchical clustering of these hub genes revealed three distinct clusters, reflecting heterogeneous expression patterns across sample groups (Fig 7B).

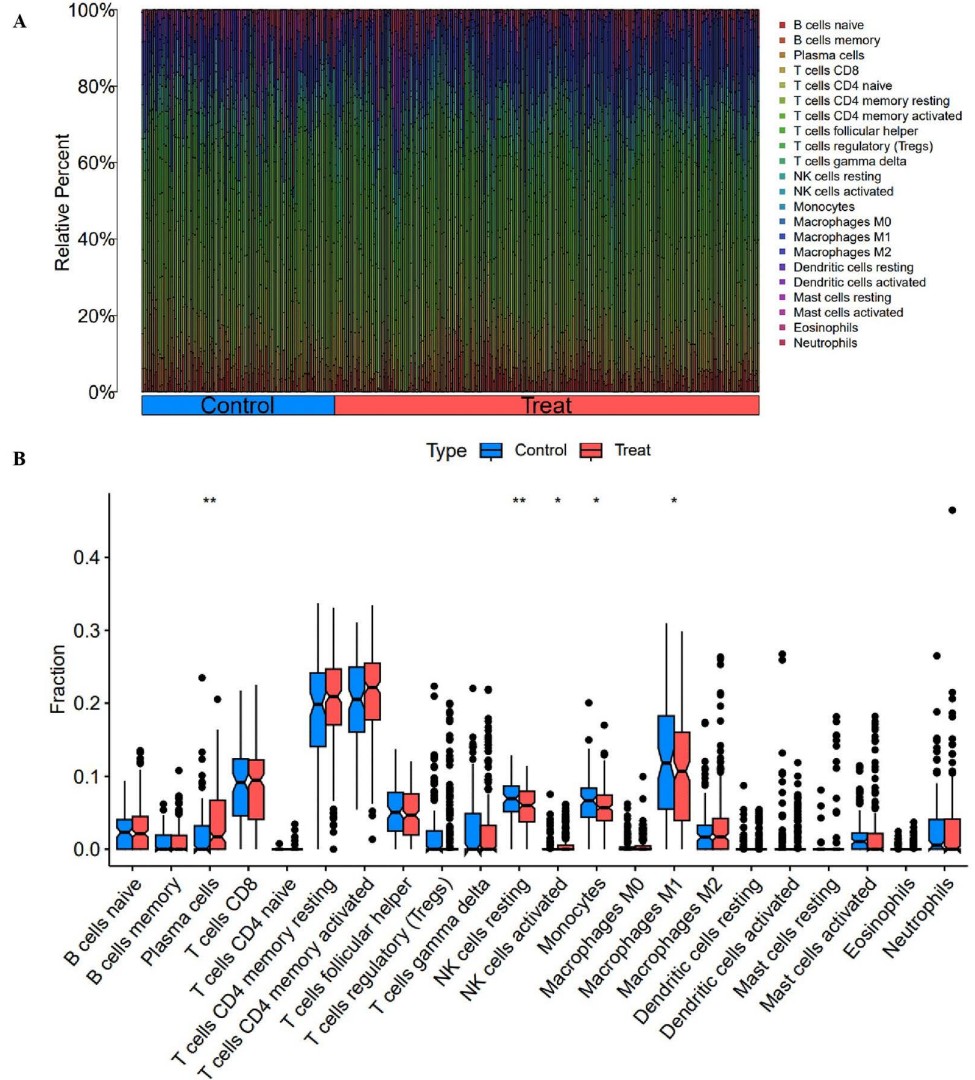

**Fig 4. Immune infiltration profiling.** Stacked bar plot showing the relative abundance of 22 immune cell subsets (estimated by CIBERSORT LM22 signature matrix) across samples in control and treatment groups, illustrating potential compositional differences in heart failure immune microenvironment.

These genes are implicated in fibrosis, contractility, and stress response pathways potentially relevant to heart failure remodeling and LBBP outcomes.

### 3.6. Drug–pathway prediction

Drug–pathway mapping using connectivity scores identified candidate compounds capable of reversing the observed transcriptional signatures. Predicted agents included β-blockers (e.g., carvedilol, metoprolol, bisoprolol, ivabradine) linked to sympathetic and remodeling pathways; RAAS inhibitors (e.g., sacubitril/valsartan, lisinopril, enalapril, losartan, valsartan) associated with hemodynamic and fibrotic regulation; anti-fibrotic agents (e.g., pirfenidone, trichostatin A) targeting extracellular matrix processes; STAT3/IL6 modulators; metabolic regulators (e.g., metformin, dapagliflozin, empagliflozin);

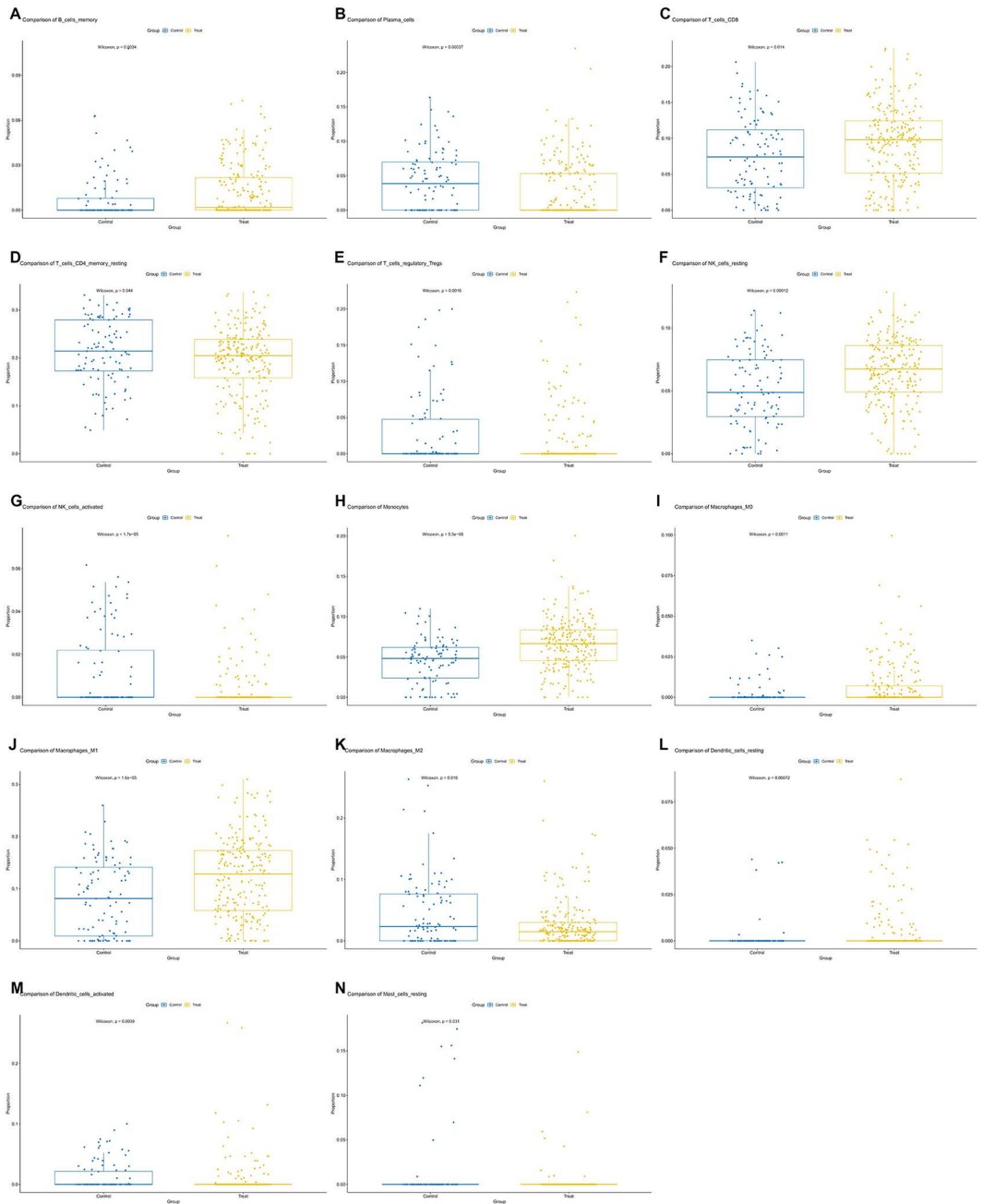

**Fig 5. Validation of immune cell differences.** Boxplots depicting significant differences (Wilcoxon rank-sum test, Benjamini-Hochberg adjusted *P<0.05, **P<0.01) in representative immune populations, including memory B cells, plasma cells, CD8 T cells, regulatory T cells, resting NK cells, monocytes, macrophage subsets (M0/M1/M2), dendritic cells, and mast cells. These alterations highlight potential immune remodeling relevant to heart failure and conduction system pacing therapies.

calcium-handling agents (e.g., digoxin, omecamtiv mecarbil); and the sGC stimulator vericiguat aligned with cGMP–PKG signaling (Fig 8). These candidates represent hypothesis-generating therapeutic options for modulating HF features potentially influencing LBBP response heterogeneity.

## Module–Trait Relationships

**Fig 6. Weighted gene co-expression network analysis (WGCNA).** Module–trait correlation heatmap showing associations between gene modules and treatment status. Significant modules (MEpink, MElightyellow, MEyellow, MEgreenyellow) exhibited robust correlations (adjusted P<0.05) with transcriptomic variation linked to remodeling processes.

## 4. Discussion

This study systematically integrated multiple transcriptomic datasets to characterize potential molecular features in HF that may relate to LBBP response heterogeneity. Through rigorous batch correction, differential gene screening, immune infiltration profiling, weighted gene co-expression network analysis (WGCNA), and drug–pathway mapping, we identified extracellular matrix remodeling, immune activation, and metabolic pathways as central mechanisms. Key genes such as ASPN, HOPX, MYH6, SERPINA3, and NPPA were prioritized, while candidate agents including β-blockers, RAAS

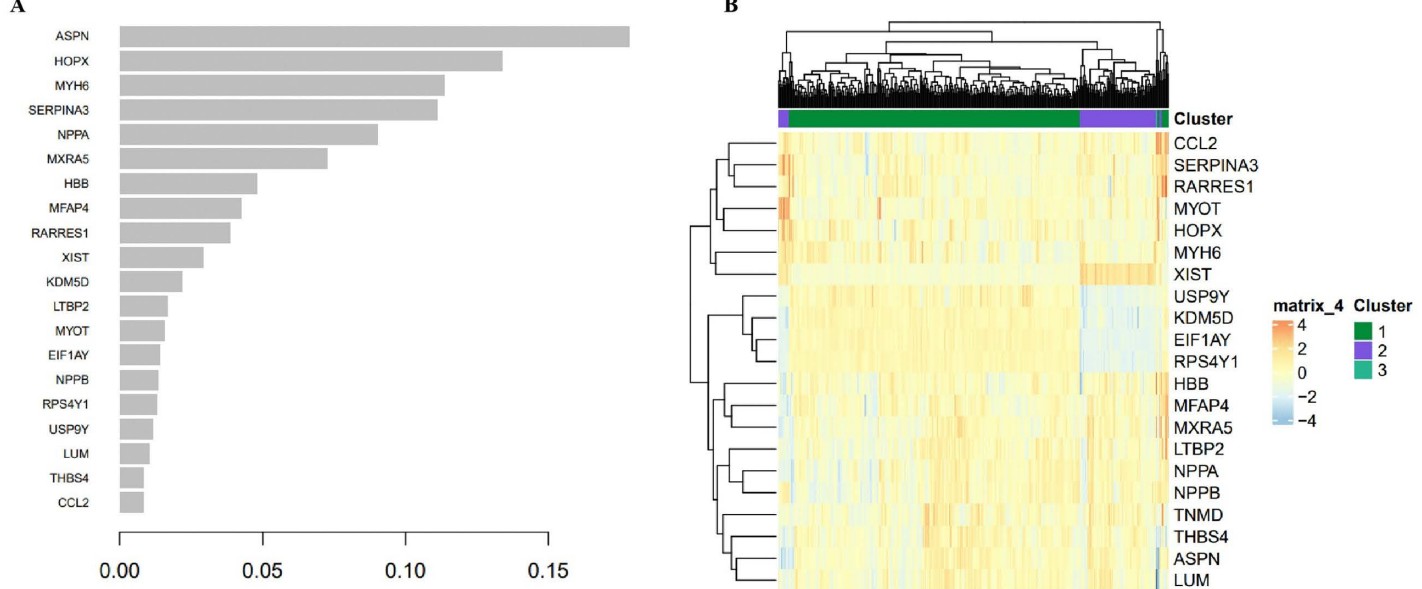

**Fig 7. Identification of hub genes.** (A) Bar plot ranking hub genes by intramodular connectivity within significant WGCNA modules, with top candidates ASPN, HOPX, MYH6, SERPINA3, and NPPA. **(B)** Heatmap and dendrogram of hub gene expression patterns across samples, clustered by treatment group, revealing heterogeneous profiles.

inhibitors, antifibrotic compounds, and SGLT2 inhibitors were suggested as potential therapeutic targets. These findings provide new hypothesis-generating insights into HF molecular mechanisms potentially influencing LBBP and propose testable therapeutic hypotheses for future validation.

Immune remodeling emerged as one of the most prominent features. Significant changes in B cell memory subsets, plasma cells, CD8 T cells, regulatory T cells, macrophage subtypes, dendritic cells, and mast cells indicate that LBBP may influence the myocardial immune microenvironment [17]. Previous studies have shown that abnormal T cell activation and macrophage polarization drive adverse ventricular remodeling in patients with heart failure [18–20]. Moreover, clinical evidence suggests that immune signatures can stratify prognosis and predict responses to device therapy in advanced cardiomyopathy [21–22]. The immune alterations observed in our analysis are therefore consistent with the emerging view that immune–inflammatory activity is a critical determinant of pacing efficacy and long-term remodeling outcomes.

Co-expression network analysis further supported these findings by identifying modules strongly associated with treatment status. Specifically, the MEpink, MElightyellow, MEyellow, and MEgreenyellow modules showed significant correlations with transcriptomic variation, suggesting their involvement in structural and functional remodeling. Hub genes such as MYH6 and NPPA are well-recognized markers of cardiac contractility and stress response [23–24], whereas ASPN and SERPINA3 have been increasingly implicated in fibrosis and extracellular matrix dysregulation in failing myocardium [25–26]. Recent transcriptomic and proteomic studies have confirmed that fibrotic remodeling is a key determinant of resynchronization outcomes [27], lending biological plausibility to our hub gene findings. Together, these results suggest that the identified genes may serve as biomarkers of treatment response or as mechanistic targets for LBBP.

Drug–pathway mapping added translational value by identifying candidate compounds capable of modulating the observed pathways. β-blockers and RAAS inhibitors remain cornerstone therapies in heart failure and have been consistently shown in large randomized trials to improve survival and reduce hospitalization. The prediction of antifibrotic drugs such as pirfenidone and trichostatin A is consistent with recent experimental studies demonstrating that inhibition

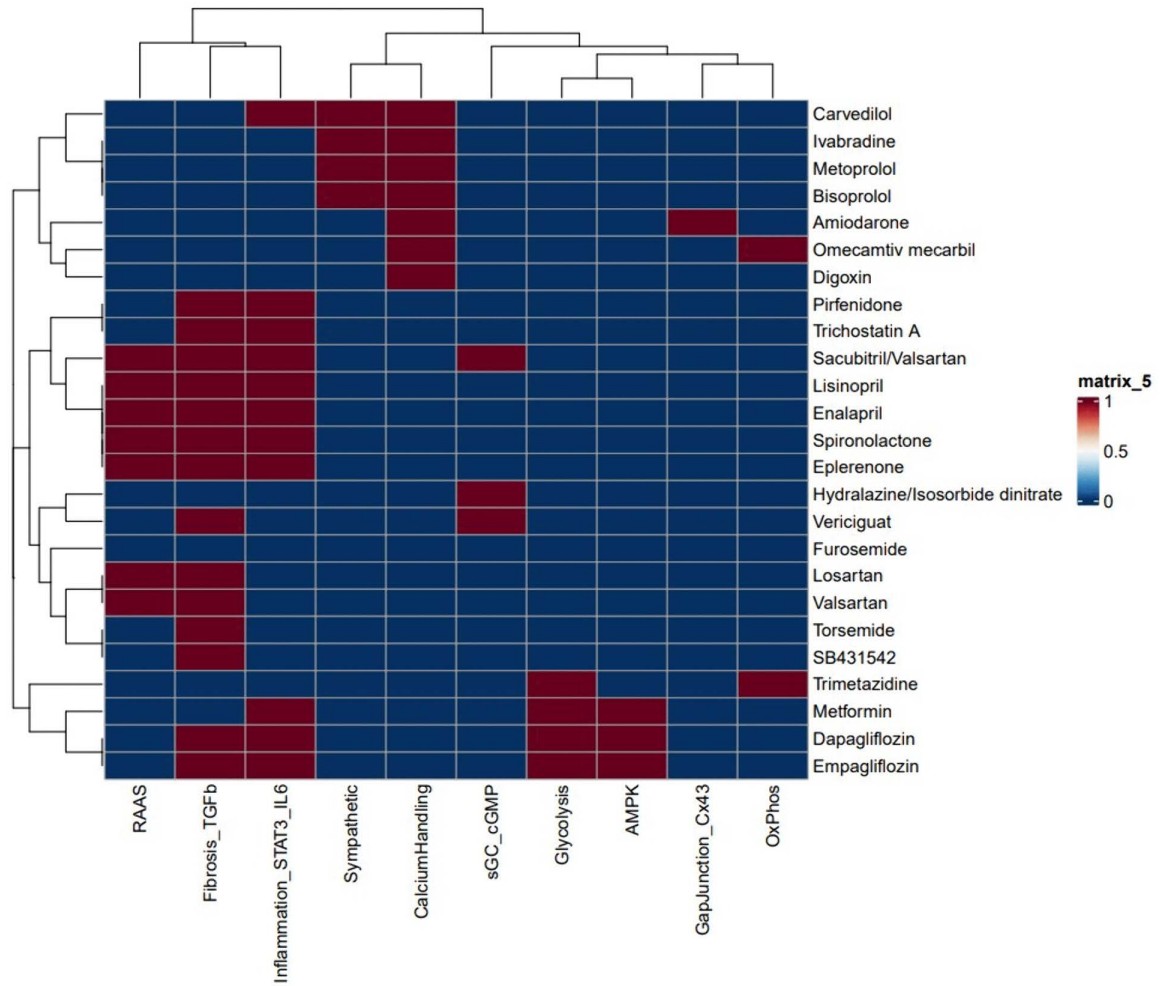

**Fig 8. Drug–pathway prediction.** Heatmap of predicted drug–pathway interactions based on connectivity mapping. Highlighted candidates include β-blockers (carvedilol, metoprolol, bisoprolol, ivabradine), RAAS inhibitors (sacubitril/valsartan, lisinopril, enalapril, losartan, valsartan), anti-fibrotic agents (pirfenidone, trichostatin A), metabolic regulators (metformin, dapagliflozin, empagliflozin), and vericiguat (sGC stimulator), aligned with enriched pathways such as cGMP–PKG and extracellular matrix remodeling.

of extracellular matrix deposition enhances the benefit of resynchronization [28–29]. Similarly, SGLT2 inhibitors, including dapagliflozin and empagliflozin, have been shown to markedly reduce cardiovascular mortality and heart failure events, and their identification in this study further underscores their role in metabolic and structural remodeling. In addition, vericiguat, a stimulator of the sGC–cGMP pathway, has demonstrated efficacy in high-risk heart failure populations, emphasizing the translational significance of our predictions.

This study provides a comprehensive systems-level perspective on HF transcriptomics potentially applicable to LBBP but several limitations should be acknowledged. First, the datasets analyzed were retrospective and heterogeneous in origin, predating the introduction of LBBP in 2017, which may introduce unmeasured confounders and limit direct applicability to LBBP-specific mechanisms. This creates a potential for circular logic, as the analyses reflect general HF biology rather than pacing-induced changes; thus, findings should be interpreted as exploratory proxies requiring validation in prospective LBBP cohorts. Second, immune infiltration results were based on computational deconvolution and require

validation using histological or flow cytometric approaches. Third, although hub genes and drug predictions were systematically derived, their clinical applicability requires further validation in cellular and animal models. Despite these limitations, the integration of transcriptomic profiling, immune landscape analysis, network biology, and drug prediction offers a novel and rigorous framework that deepens the mechanistic understanding of HF remodeling potentially relevant to LBBP and lays a foundation for future translational research.

## 5. Conclusion

This integrative transcriptomic analysis of HF profiles revealed potential associations with extracellular matrix remodeling, immune cell alterations, and key hub genes including ASPN, HOPX, MYH6, SERPINA3, and NPPA that may influence LBBP response heterogeneity. Network-based approaches identified modules significantly linked to treatment response, and drug–pathway mapping highlighted candidate agents such as β-blockers, RAAS inhibitors, antifibrotic compounds, SGLT2 inhibitors, and vericiguat. These findings provide hypothesis-generating insights into HF mechanisms relevant to LBBP and suggest potential therapeutic strategies for improving clinical outcomes in dedicated validation studies.

## Supporting information

**S1 Data. Raw transcriptomic data.**
(RAR)

## Author contributions

**Conceptualization:** Mingfeng Jin.

**Data curation:** Wei Yuan.

**Formal analysis:** Xiang Tang.

**Methodology:** Mingfeng Jin.

**Writing – original draft:** Xia Sun, Wei Zhong.

**Writing – review & editing:** Xia Sun, Wei Zhong.

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
