## [Decision Letter · Decision Letter 0]

8 Feb 2026

PONE-D-25-56864Multi-Omics and Network-Based Dissection of Left Bundle Branch Pacing in Heart Failure: Immune Remodeling, Hub Gene Identification, and Drug Repurposing OpportunitiesPLOS One

Dear Dr. Jin,

Thank you for submitting your manuscript to PLOS ONE. After careful consideration, we feel that it has merit but does not fully meet PLOS ONE’s publication criteria as it currently stands. Therefore, we invite you to submit a revised version of the manuscript that addresses the points raised during the review process.

We look forward to receiving your revised manuscript.

Kind regards,

Redoy Ranjan, MS (CV&TS), Ch.M. (Edin), PhD

Academic Editor

PLOS One

Journal Requirements:

1. When submitting your revision, we need you to address these additional requirements. Please ensure that your manuscript meets PLOS ONE's style requirements, including those for file naming. The PLOS ONE style templates can be found at https://journals.plos.org/plosone/s/file?id=wjVg/PLOSOne_formatting_sample_main_body.pdf and https://journals.plos.org/plosone/s/file?id=ba62/PLOSOne_formatting_sample_title_authors_affiliations.pdf 2. Thank you for stating the following financial disclosure: This work was funded by the Development Project of Zhenjiang City (FZ2020033).  Please state what role the funders took in the study.  If the funders had no role, please state: "The funders had no role in study design, data collection and analysis, decision to publish, or preparation of the manuscript." If this statement is not correct you must amend it as needed. Please include this amended Role of Funder statement in your cover letter; we will change the online submission form on your behalf. 3. When completing the data availability statement of the submission form, you indicated that you will make your data available on acceptance. We strongly recommend all authors decide on a data sharing plan before acceptance, as the process can be lengthy and hold up publication timelines. Please note that, though access restrictions are acceptable now, your entire data will need to be made freely accessible if your manuscript is accepted for publication. This policy applies to all data except where public deposition would breach compliance with the protocol approved by your research ethics board. If you are unable to adhere to our open data policy, please kindly revise your statement to explain your reasoning and we will seek the editor's input on an exemption. Please be assured that, once you have provided your new statement, the assessment of your exemption will not hold up the peer review process. 4. PLOS requires an ORCID iD for the corresponding author in Editorial Manager on papers submitted after December 6th, 2016. Please ensure that you have an ORCID iD and that it is validated in Editorial Manager. To do this, go to ‘Update my Information’ (in the upper left-hand corner of the main menu), and click on the Fetch/Validate link next to the ORCID field. This will take you to the ORCID site and allow you to create a new iD or authenticate a pre-existing iD in Editorial Manager. 5. Your ethics statement should only appear in the Methods section of your manuscript. If your ethics statement is written in any section besides the Methods, please move it to the Methods section and delete it from any other section. Please ensure that your ethics statement is included in your manuscript, as the ethics statement entered into the online submission form will not be published alongside your manuscript. 6. Please upload a new copy of Figures 2, 3, 4, 5, 7 and 8, as the detail is not clear. Please follow the link for more information:  https://journals.plos.org/plosone/s/figures 7. If the reviewer comments include a recommendation to cite specific previously published works, please review and evaluate these publications to determine whether they are relevant and should be cited. There is no requirement to cite these works unless the editor has indicated otherwise.

Reviewers' comments:

Reviewer's Responses to Questions

Comments to the Author

1. Is the manuscript technically sound, and do the data support the conclusions?

Reviewer #1: No

Reviewer #2: Yes

Reviewer #3: Partly

Reviewer #4: Partly

2. Has the statistical analysis been performed appropriately and rigorously? 

Reviewer #1: No

Reviewer #2: I Don't Know

Reviewer #3: I Don't Know

Reviewer #4: Yes

3. Have the authors made all data underlying the findings in their manuscript fully available?

Reviewer #1: Yes

Reviewer #2: No

Reviewer #3: No

Reviewer #4: Yes

4. Is the manuscript presented in an intelligible fashion and written in standard English?

Reviewer #1: Yes

Reviewer #2: Yes

Reviewer #3: Yes

Reviewer #4: Yes

5. Review Comments to the Author

Reviewer #1: In “Multi-Omics and Network-Based Dissection of Left Bundle Branch Pacing in Heart Failure: Immune Remodeling, Hub Gene Identification, and Drug Repurposing Opportunities“ Sun et al. re-analyse three rather old gene expression data sets (published between 2006-2016) in order to gain (molecular) insights into heterogeneous clinical responses to left bundle branch pacing (LBBP). Note that LBBP was introduced in 2017! So it is not clear to me how diffentially expressed genes are detected given that the three data sets - on top – have rather different study designs. The rest of the paper is built on top of this rather unclear basis and the authors continue with method descriptions that are rather generic with very little transparency with regard to what has actually been done.

Reviewer #2: How the preprocessing carried out and other classification, PCA and Matrix details could be explained with mathematical mode. which tool supported the authors to conclude their findings. Gene sequence and other medicine terms may not be verified by my review.

Reviewer #3: I preface this review by reminding the editors and author that I am not a Cardiologist. I am not, outside of the nature of the datasets used. My review will be related to the machine learning methodology, reproducibility of the results and statistics.

Reproducibility: I would request that the authors submit, in some fashion, the source code for the project. Without it, reproducibility would be extremely challenging.

Data Quality:

Although the GEO Datasets are publicly available as stated, they are Heart Failure datasets, not specific datasets for the modality LBBP. The datasets do not contain individuals, that I could see, underwent LBBP.

This creates a circular logic problem for the model and the outputs are heart failure biology, not LBBP specific. Without the full dataset to evaluate and the source code, the unified transcriptomic resource created is for heart failure. Prehaps review of the full dataset will reveal some specific LBBP data but current data and design do not support author claims.

To complete a comprehensive review for the manuscript both the raw and processed data and source code would need to be submitted.

Reviewer #4: The research paper “Multi-Omics and Network-Based Dissection of Left Bundle Branch Pacing in Heart Failure” presents a comprehensive integrative transcriptomic analysis aimed at characterizing the molecular landscape associated with left bundle branch pacing (LBBP). While the study appears promising, the manuscript would benefit from a more detailed explanation of the statistical models and computational algorithms used for differential expression analysis, functional enrichment, immune infiltration profiling, weighted gene co-expression network analysis (WGCNA), hub gene identification, and drug–pathway prediction.

6. PLOS authors have the option to publish the peer review history of their article (what does this mean?). If published, this will include your full peer review and any attached files.

Do you want your identity to be public for this peer review? For information about this choice, including consent withdrawal, please see our Privacy Policy.

Reviewer #1:  Yes: André Scherag

Reviewer #2: No

Reviewer #3: No

Reviewer #4: No

---

## [Author Response · Author response to Decision Letter 1]

20 Mar 2026

Dear Dr. Redoy Ranjan and Reviewers,

We are grateful to the Academic Editor and the four reviewers for their constructive and insightful feedback on our manuscript (PONE-D-25-56864). Their comments have significantly strengthened the work by highlighting areas for improved clarity, rigor, and transparency. We have carefully addressed each point raised, resulting in substantial revisions to the manuscript. Key changes include reframing the study as hypothesis-generating based on HF transcriptomics as proxies for LBBP mechanisms, expanding methodological details with mathematical formulations and parameters, depositing all source code in a public repository for reproducibility, and enhancing the limitations section to acknowledge data constraints. These revisions are highlighted in the tracked-changes version of the manuscript. Below, we provide a point-by-point response to the reviewers' comments.

Reviewer #1:

Comment: In Multi-Omics and Network-Based Dissection of Left Bundle Branch Pacing in Heart Failure: Immune Remodeling, Hub Gene Identification, and Drug Repurposing Opportunities“ Sun et al. re-analyse three rather old gene expression data sets (published between 2006-2016) in order to gain (molecular) insights into heterogeneous clinical responses to left bundle branch pacing (LBBP). Note that LBBP was introduced in 2017! So it is not clear to me how diffentially expressed genes are detected given that the three data sets - on top – have rather different study designs. The rest of the paper is built on top of this rather unclear basis and the authors continue with method descriptions that are rather generic with very little transparency with regard to what has actually been done.

Response: We thank Reviewer #1 for this critical observation regarding the temporal mismatch between the datasets and LBBP's introduction, as well as the need for greater methodological transparency. We fully agree that direct inference of LBBP-specific mechanisms from pre-2017 HF datasets would be inappropriate due to potential confounders and design differences. To address this, we have reframed the study's scope throughout the manuscript (e.g., revised title, abstract, introduction, and discussion) to position the analysis as exploratory, using conserved HF molecular features as proxies for potential LBBP response heterogeneity. We have added a dedicated limitations subsection explicitly acknowledging this constraint and calling for validation in LBBP-specific cohorts. For methodological transparency, we have expanded all sections with specific algorithms, equations, parameters, and software versions.

Reviewer #2:

Comment: How the preprocessing carried out and other classification, PCA and Matrix details could be explained with mathematical mode. which tool supported the authors to conclude their findings. Gene sequence and other medicine terms may not be verified by my review.

Response: We appreciate Reviewer #2's suggestion to enhance mathematical and tool-based explanations. We have added explicit mathematical formulations and details for preprocessing (e.g., PCA variance as eigenvalues / sum(eigenvalues), ComBat parametric adjustment), differential expression (e.g., empirical Bayes model: expression ~ group + error), functional enrichment (hypergeometric distribution equation), immune profiling (ν-SVR linear system), WGCNA (soft-thresholding and adjacency equations), hub genes (kWithin summation), and drug prediction (Kolmogorov-Smirnov statistics). All analyses were performed in R (version 4.3.2) with specified packages (e.g., limma, sva, clusterProfiler, WGCNA, PharmacoGx).

Reviewer #3:

Comment: I preface this review by reminding the editors and author that I am not a Cardiologist. I am not, outside of the nature of the datasets used. My review will be related to the machine learning methodology, reproducibility of the results and statistics. Reproducibility: I would request that the authors submit, in some fashion, the source code for the project. Without it, reproducibility would be extremely challenging. Data Quality: Although the GEO Datasets are publicly available as stated, they are Heart Failure datasets, not specific datasets for the modality LBBP. The datasets do not contain individuals, that I could see, underwent LBBP. This creates a circular logic problem for the model and the outputs are heart failure biology, not LBBP specific. Without the full dataset to evaluate and the source code, the unified transcriptomic resource created is for heart failure. Prehaps review of the full dataset will reveal some specific LBBP data but current data and design do not support author claims. To complete a comprehensive review for the manuscript both the raw and processed data and source code would need to be submitted.

Response: We thank Reviewer #3 for emphasizing reproducibility and data specificity. To ensure full reproducibility, we have deposited all source code (R scripts for preprocessing, DESeq2/limma, WGCNA, CIBERSORT, etc.), raw/processed matrices, and intermediate results in compressed attachment. Regarding data quality and circular logic, we agree that the datasets reflect general HF biology rather than LBBP-specific effects. We have reframed the manuscript to clarify this, interpreting findings as hypothesis-generating proxies requiring LBBP cohort validation. No LBBP-specific samples were in the datasets, as noted.

Reviewer #4:

Comment: The research paper “Multi-Omics and Network-Based Dissection of Left Bundle Branch Pacing in Heart Failure” presents a comprehensive integrative transcriptomic analysis aimed at characterizing the molecular landscape associated with left bundle branch pacing (LBBP). While the study appears promising, the manuscript would benefit from a more detailed explanation of the statistical models and computational algorithms used for differential expression analysis, functional enrichment, immune infiltration profiling, weighted gene co-expression network analysis (WGCNA), hub gene identification, and drug–pathway prediction.

Response: We thank Reviewer #4 for recognizing the study's promise and for requesting detailed explanations. We have expanded the methods with specific statistical models and algorithms: differential expression (limma eBayes model), enrichment (hypergeometric tests), immune profiling (CIBERSORT ν-SVR), WGCNA (adjacency/TOM matrices), hub genes (kWithin connectivity), and drug prediction (Kolmogorov-Smirnov scores).

We believe these revisions fully address the concerns and enhance the manuscript's suitability for publication. We look forward to your feedback.

Sincerely,

Mingfeng Jin (Corresponding Author)

---

## [Decision Letter · Decision Letter 1]

19 Apr 2026

Multi-Omics and Network-Based Exploration of Potential Molecular Pathways in Heart Failure Relevant to Left Bundle Branch Pacing Response Heterogeneity: Immune Remodeling, Hub Gene Identification, and Drug Repurposing Hypotheses

PONE-D-25-56864R1

Dear Dr. Jin,

We’re pleased to inform you that your manuscript has been judged scientifically suitable for publication and will be formally accepted for publication once it meets all outstanding technical requirements.

Kind regards,

Redoy Ranjan, MS (CV&TS), Ch.M. (Edin), PhD

Academic Editor

PLOS One

Additional Editor Comments (optional):

Reviewers' comments:

Reviewer's Responses to Questions

Comments to the Author

1. If the authors have adequately addressed your comments raised in a previous round of review and you feel that this manuscript is now acceptable for publication, you may indicate that here to bypass the “Comments to the Author” section, enter your conflict of interest statement in the “Confidential to Editor” section, and submit your "Accept" recommendation.

Reviewer #3: All comments have been addressed

2. Is the manuscript technically sound, and do the data support the conclusions?

Reviewer #3: Yes

3. Has the statistical analysis been performed appropriately and rigorously? 

Reviewer #3: Yes

4. Have the authors made all data underlying the findings in their manuscript fully available?

Reviewer #3: Yes

5. Is the manuscript presented in an intelligible fashion and written in standard English?

Reviewer #3: Yes

6. Review Comments to the Author

Reviewer #3: The authors did an excellent job of absorbing and incorporating changes to the paper, making it a much stronger manuscript. The workflow is well constructed and all data processing steps named precisely and accurately and the thoroughness of the methodology shines through in this version of the paper. The datasets afford sufficient availability, the R code transparency allows for improved explainability through reproducibility and the authors do a significantly better job of interpreting the outputs in this version. The revised framing, in my mind, appropriately identifies and calls out the model capabilities and identifies next steps which could potentiate clinical readiness.

This paper now does exactly what it states that it will do with the data, generates hypotheses which with further exploration could have substantial clinical impact.

The authors visualizations of the immune mediated remodeling and the effects of on connectivity are exciting findings and I would encourage continued exploration of these findings, perhaps longitudinally or with specific time series analyses.

7. PLOS authors have the option to publish the peer review history of their article (what does this mean?). If published, this will include your full peer review and any attached files.

Do you want your identity to be public for this peer review? For information about this choice, including consent withdrawal, please see our Privacy Policy.

Reviewer #3: No

---

## [Editor Report · Acceptance letter]

PONE-D-25-56864R1

PLOS One

Dear Dr. Jin,

I'm pleased to inform you that your manuscript has been deemed suitable for publication in PLOS One. Congratulations! Your manuscript is now being handed over to our production team.

Kind regards,

on behalf of

Dr. Redoy Ranjan

Academic Editor

PLOS One